# Vitamin K Epoxide Reductase Complex–Protein Disulphide Isomerase Assemblies in the Thiol–Disulphide Exchange Reactions: Portrayal of Precursor-to-Successor Complexes

**DOI:** 10.3390/ijms25084135

**Published:** 2024-04-09

**Authors:** Maxim Stolyarchuk, Marina Botnari, Luba Tchertanov

**Affiliations:** Centre Borelli, ENS Paris-Saclay, CNRS, Université Paris-Saclay, 4 Avenue des Sciences, 91190 Gif-sur-Yvette, France; maxim.stolyarchuk@gmail.com (M.S.); botnarimarina97@gmail.com (M.B.)

**Keywords:** PDI–hVKORC1, thiol–disulphide exchange reactions, molecular interactions, transient states, precursor, intermediate, and successor complexes, 3D modelling, molecular dynamics, computational biophysics, allosteric regulation, protein folding, intrinsic disorder

## Abstract

The human Vitamin K Epoxide Reductase Complex (hVKORC1), a key enzyme that converts vitamin K into the form necessary for blood clotting, requires for its activation the reducing equivalents supplied by its redox partner through thiol–disulphide exchange reactions. The functionally related molecular complexes assembled during this process have never been described, except for a proposed de novo model of a ‘precursor’ complex of hVKORC1 associated with protein disulphide isomerase (PDI). Using numerical approaches (*in silico* modelling and molecular dynamics simulation), we generated alternative 3D models for each molecular complex bonded either covalently or non-covalently. These models differ in the orientation of the PDI relative to hVKORC1 and in the cysteine residue involved in forming protein–protein disulphide bonds. Based on a comparative analysis of these models’ shape, folding, and conformational dynamics, the most probable putative complexes, mimicking the ‘precursor’, ‘intermediate’, and ‘successor’ states, were suggested. In addition, we propose using these complexes to develop the ‘*allo-network drugs*’ necessary for treating blood diseases.

## 1. Introduction

Human vitamin K epoxide reductase complex 1 (hVKORC1), an intrinsic endoplasmic reticulum (ER) protein, maintains the activity of vitamin K-dependent proteins (VKDPs), which are involved in various physiological processes such as blood coagulation, cardiovascular health, and signalling in cells [1,2,3]. hVKORC1 is the target of vitamin K antagonists (VKAs), the membrane-impermeable anticoagulant drugs commonly used for the prevention and treatment of various diseases that are associated with the deregulated activity of hVKORC1 [4].

As an enzyme, hVKORC1 reduces a vitamin K epoxide in hydroquinone and further in quinone, the cofactor used for PKDP activation. The vitamin K transformation occurs in the active site of hVKORC1 which contains a cysteine-based CXXC motif, highly conserved among hVKORC1 homologues from plants, bacteria, archaea, and mammals [5].

To achieve accurate vitamin K transformation, hVKORC1 needs to be activated regularly, and this activation process is a less studied step in the hVKORC1 life cycle. Although various hypotheses have been proposed based on biochemical, biophysical, *in silico*, and crystallographic studies, the explicit mechanisms of hVKORC1 activation required to maintain vitamin K levels are currently undefined. It is generally accepted that two conserved cysteine residues located in the luminal loop (L-loop) are involved in biochemical reactions between hVKORC1 and its redox protein, which delivers the required reducing equivalents to control the hVKORC1 activity [6,7].

In total, three types of biochemical reactions are responsible for chemical transformations between the functional groups of such proteins: nucleophilic substitution–elimination, specifically so-called thiol–disulphide exchange (TDSE) reactions, two-electron reactions of thiols or disulphide bonds, as well as radical reactions [8]. These reactions promote the transformations of the disulphide bond (–S-S–), thiol (–SH), and thiolate anion (–S–) and therefore play a central role in the oxidative folding of proteins and are a key mechanism in almost all enzymes that form and isomerise disulphide bonds [9]. During these reactions, one of the thiol groups of the redox protein is deprotonated and carries out a nucleophilic attack on the substrate (target) disulphide bond, forming a mixed disulphide intermediate (Figure 1A). Another thiol group releases the reduced substrate (target), leaving behind the oxidised enzyme.

The nucleophilic attack of a thiolate on a disulphide is directed along the disulphide axis, and this requirement for collinearity establishes the orientation necessary for interactions between redox partners [7,10]. Thus, the TDSE reaction has significant steric requirements that must be met by enzymes capable of adapting their folding at each process step.

In the hVKORC1 oxidation–reduction cycle, the initial step is triggered by the transfer of electrons from two thiol groups (–SH) of a redox protein to the –S-S– bond in the L-loop of hVKORC1, producing the reduced (activated) state of hVKORC1 (Figure 1B). Through this mechanism, the –S-S– disulphide bond formed by C43 and C51 of hVKORC1 (substrate, target) is exchanged with the pair of thiol groups from C37 and C40 of PDI (redox, donor) [5,11]. The second step is to transfer these electrons to the CXXC motif of the hVKORC1 enzymatic active site to reduce its cysteine residues (the intramolecular TDSE reactions). Finally, the protonated CXXC motif of hVKORC1 uses electron/proton transfer to transform vitamin K into quinone, which leaves the oxidised enzyme [12]. Therefore, each step of vitamin K transformation is closely linked to the reduction of the CXXC motif in the hVKORC1 active site, which in turn depends on the protonation of L-loop cysteine residues by the redox protein [13]. To maintain the reduced vitamin K level, VKORC1 must be regularly activated by a redox protein.

Identification of the physiological redox partner of hVKORC1 has been the subject of numerous in vitro and in vivo studies for an extended period and is still being investigated [14,15]. The hVKORC1-specific redox protein was predicted to be either the protein disulphide isomerase (PDI) or one of the thioredoxin-like proteins—thioredoxin-related transmembrane proteins (Tmx1 and Tmx4) and endoplasmic reticulum oxidoreductase (ERp18). We recently proposed, using in silico methods, that PDI is the protein most compatible with hVKORC1 compared to the thioredoxin-like proteins and is most likely its redox protein [16]. Later, our prediction was confirmed by an empirical study demonstrating the functional interaction between VKORC1 and PDI [17].

Using knowledge-based arguments, two models of the precursor PDI–hVKORC1 molecular complex, differing by alternative positions of the PDI with respect to hVKORC1, were postulated [16]. These models were successfully reproduced by the automatic protein–protein docking [18]. In both PDI–hVKORC1 complex modelling approaches, the de novo model of hVKORC1 in the oxidised (inactive) state was used as a target of PDI [19]. This *de novo* model of hVKORC1, validated by crystallographic data [2], comprises the stable transmembrane four-helix bundle crowned by the intrinsically disordered (ID) luminal loop (L-loop, R33-N77), protruding in the ER lumen. In both alternative PDI–hVKORC1 complex models, proteins bind to each other using a combination of hydrogen bonds, salt bridges, and hydrophobic contacts formed by residues from the different protein domains [16]. The interatomic contacts between sulphur atoms from cysteine residues of PDI and hVKORC1 in both models are adequate for a study of the TDSE reactions between these two proteins.

To advance our knowledge of the dynamical synergy of hVKORC1 with its redox protein PDI during the initial step of hVKORC1 activation, we focus on modelling PDI–hVKORC1 molecular complexes assembled during the TDSE reactions. Since this process represents a less studied step of the hVKORC1 life cycle, such modelling will provide novel insight into hVKORC1 functions and open perspectives for future research. Thus, establishing the 3D structures of ‘precursor’, ‘intermediate’, and ‘successor’ complexes may represent an important biophysics element for studying intermolecular TDSE reactions and could be included in quantitative analyses of the diverse dynamics of the PDI-hVKORC1 system as well as its connection with functional properties.

We report here (i) an extended analysis of the PDI–hVKORC1 ‘precursor’ complex, formed prior to proton–electron transfer (state I), and (ii) the modelling of two other complexes assembled during TDSE reactions—a covalently bound ‘intermediate’ complex, chemically linking two proteins (state II), and a non-covalently bound ‘successor’ complex (state III) (Figure 1B).

An ‘intermediate’ state of the PDI–hVKORC1 complex is stabilised by the covalent ‘mixed’ –S-S– bond between the thiol groups of one of the flanking cysteine residues from the CGHC motif of PDI, implementing a nucleophilic attack on the sulphur atom of the C43-C51 disulphide bond of hVKORC1. When a proton of the second thiol group of PDI moves towards hVKORC1, and locks at its thiolate (–S–), the two proteins separate from each other, forming the final ‘successor’ PDI-hVKORC1 non-covalent complex, leading to the disassembly of the two proteins.

## 2. Results

### 2.1. Modelling and Data Processing

Structural models of the PDI-hVKORC1 ‘precursor’ complex, Models I-1 and I-2, describing the initial state before TDSE reactions (state I), were taken from [16] and utilised for more extended (0.52 μs) MD simulations. Using the MD conformations of the PDI-hVKORC1 precursor complex as templates, other models of PDI-hVKORC1 forming during TDSE reactions were constructed by converting thiol groups to a disulphide bond and vice versa. Because the thiol group of PDI can interact with either C41 or C53 of hVKORC1 to form a ‘mixed’ –S-S– bond, the modelling of the ‘intermediate’ complex (state II) is represented by a pair of models derived from each model of the ‘precursor ‘complex, Models I-1 and I-2. Generated models of the ‘intermediate’ complex are labelled Models II-11, II-12, II-21, and II-22. The ‘successor’ complex describing the final state after TDSE reactions is referred to as Model III-1.

All constructed 3D models were inserted into the membrane (Figure 2A) and explored by conventional MD simulations (all atoms, explicit water, trajectory of 0.52 μs). During the first step (0–0.18 µs), each model was simulated using the constraints for the S⋯S inter-protein distance, which were further relaxed. The MD conformations of each model were normalised by least-squares fitting to the reference structure (the conformations taken at t = 0.18 μs), to avoid the motion of the protein as a rigid body. To compare models, we used the same protocol and criteria for all PDI-hVKORC1 complex constructions, their MD simulation, and the analysis of generated data as described in [16].

### 2.2. The ‘Precursor’ PDI-hVKORC1 Complex: Which De Novo Model Is More Appropriate?

Since experimental confirmation of a de novo model of the ‘precursor’ complex PDI-hVKORC1 assembled prior to the TDSE reactions (state I) has yet to be obtained, we decided to examine these putative models in more detail.

Model I-1 and Model I-2, which differ in the orientation of PDI relative to hVKORC1, were simulated under conditions mimicking the natural environment to explore the stability of each system and determine the distance between cysteine residues from two proteins, PDI and hVKORC1, which is optimal for proton transfer (Figure 2B). The MD simulation showed that Model I-1 is more stable compared to Model I-2 as its conformations exhibit a more conservative geometry (molecular complex shape, folding, and conformation of each protein partner) maintained by electrostatic interactions of R61 (hVKORC1) with E46 (PDI) through salt bridge formation and with A42 (PDI), forming a strong and stable H-bond (Figure 2C–F and Appendix A). Such electrostatic interactions, together with van der Waals contacts, stabilise the close spatial position of the two proteins with an optimal distance between the sulphur atoms, which allows the formation of an intermolecular covalent –S-S– bond.

Even if destabilised for a limited time, these non-covalent contacts are reversibly restored. Curiously, contacts of R61 with E46 and A42 break and restore at the same time. This synchronised effect may occur due to a change in the conformation of the R61 side chain. In Model I-2, no conserved contacts exist, which allows for maintaining the stability of the precursor complex PDI-hVKORC1 for at least 50–70 ns. As a result, after relaxing the constraints on the intermolecular distance S⋯S that held the simulated system, the two proteins separate and end up at a very large distance.

It is important to note that if the sulphur atoms of PDI and hVKORC1 are located at a distance of 7–8 Å, then the thiol proton (–S-H) of PDI is directed towards the –S-S– bond of hVKORC1. These conditions are favourable for the initiation of the TDSE process. If this distance reaches minimal values (5.4–6.6 Å), repulsive forces (steric and electrostatic) begin to act, contributing to a change in the orientation of the thiol proton to the side and a sharp increase in the distance between the sulphur atoms of PDI and hVKORC1. According to statistics obtained using protein structures from the Protein Database (PDB) [20], if the distance between two thiols capable of forming a disulphide bond reaches 6.2–6.6 Å, there is a high probability that a disulphide will be formed [21] (Appendix A).

The results obtained from the more extended MD simulation (0.52 μs) of the fully relaxed systems (Models I-1 and I-2) are consistent with those observed during the limited MD simulation (0.08 μs) of these models with the S⋯S distance gradually decreasing from 12 to 8 Å and used as the constrained contact [16]. The new results further support our earlier conclusion that Model I-1 most adequately represents the PDI-hVKORC1 ‘precursor’ complex. Therefore, to model the ‘intermediate complex’, we first use Model I-1.

### 2.3. The ‘Intermediate’ Covalently Bonded PDI-hVKORC1 Complex 

In this step of modelling the PDI-hVKORC1 complexes, two hypotheses were proposed: (i) the most solvent-exposed thiol group from the CGHC motif of PDI is the first to participate as a group donor in TDSE reactions; (ii) both sulphur atoms from the cysteine residues of L-loop are equally likely to be the first target atoms.

Since C37 is the most exposed residue of the PDI motif CGHC, its thiol group can interact with either C41 or C53 of hVKORC1 to form a ‘mixed’ inter-protein –S-S– bond. Therefore, the ‘intermediate’ complex (state II) was modelled considering these two possible modes of covalent binding. First, a pair of ‘intermediate’ covalent complex models was derived from the precursor complex, Model I-1, labelled as Model II-11 and Model II-12 (Figure 3).

The root-mean-square deviations (RMSDs) of the MD conformations from the initial structure (t = 0 µs) calculated for the Ca-atoms of each model of the PDI-hVKORC1 complex showed homogeneous and small RMSD values that quickly reached a plateau, indicating the quasi-equilibrium characteristics of Model II-12. Although the RMSD values of Model II-11 are three times greater than those of Model II-12, after 0.15 µs, they stabilise and, even after 0.45 µs, they begin to decrease. The RMSDs calculated for only the Cα-atoms of the L-loop show a similar tendency, but their values are twice as small as in the PDI-hVKORC1 complex in all studied models. These observations may indicate slight conformational variations in the L-loop but a displacement of PDI relative to VKORC1.

Comparing the folding of the generated MD conformations, we note that the αH1-L helix of the L-loop is well conserved in Model II-12, while it rapidly transits into the reversible 3_10_- and π-helices (Appendix A). The double helix H2-L and the short helix H3-L are transient in both models and demonstrate the reversible transition between α-3_10_- and π-helices (α ↔ 3_10_ ↔ π). Frequently the helical folding transits to bent or coil structures. Overall, the L-loop helical fold is slightly increased (by 10%) in Model II-12 compared to Model II-11. On the other hand, in PDI, the N-extremity of the αH2 helix containing the CGHC motif, in Model II-12, transits to the 3_10_-helix and then into a turn, whereas in Model II-11, this helix is well conserved in length and type of helical fold.

Further, we generated 3D models of the ‘intermediate’ complex using Model I-2 as a starting structure. These models, Model II-21 and Model II-22, were simulated and compared with Model II-11 and Model II-12 (Figure 4).

The dynamic shape of each model, represented by superimposed MD conformations, serves as a confident guide to assessing the stability and plasticity of the model, indicating its plausibility as a functionally related entity. Regarding the four alternative models, possible for the ‘intermediate’ state of the PDI-hVKORC1 complex, Model II-12 appears to fit these qualities best. This model describes an ‘intermediate’ state of the PDI-hVKORC1 complex, in which the PDI cysteine residue C37 binds to C51 of hVKORC1, forming a ‘mixed’ covalently bound complex. In this complex, the PDI changes little in its fold and overall tertiary structure. The intrinsically disordered L-loop of hVKORC1 slightly increases its helical fold and adapts its conformation to accommodate PDI. This effect is confirmed by the distance between Cα-atoms from R61 and A42, which varies slightly around an average value of 5.4 Å. The distance characterising the formation of a salt bridge between R61 and E46 gradually decreases. Both metrics indicate the good stability of inter-protein electrostatic interactions, salt bridge, and H-bonding, which further stabilise the covalently bonded molecular complex.

### 2.4. The ‘Successor’ PDI-hVKORC1 Complex

The ‘successor’ complex is a final state (state III) assembled after TDSE reactions. This state is instead described as a complex preceded by the complete dissociation of two proteins, PDI and hVKORC1. The most reliable models, Model I-1 and its intermediate derivative Model II-12, were used to model the ‘successor’ complex of PDI-hVKORC1 (Model III-1) (Figure 3). The molecular complex in state III is maintained by non-covalent interactions formed by residue R61 with A42 and E46, which maintained two proteins in close proximity as evidenced by the distance between the sulphur atoms from C43 (hVKORC1) and C37 (PDI), which is shorter overall compared to Model I-1 (Appendix A). To compare this distance between two non-covalently bound complexes, the concatenated data were normalised by fitting to the same conformation and used to calculate this metric. We found that in both models, despite the almost equal distances between the Cα-atoms of C37 and C43, their sulphur atoms can be located at both large and very close distances (Appendix A). Interestingly, short distances are observed in Model I-1 at the beginning of the MD simulation and in Model III-1 at the end of the simulation.

To estimate the binding free energy maintaining the ‘precursor’ and ‘successor’ non-covalent complexes, we used the MM-GBSA method [22]. The average energy values of −16.7 and −9.2 kcal/mol were found for Model I-1 and Model III-1, respectively. These values suggest that the weaker non-covalent interactions form the ‘successor’ complex compared to those in the ‘precursor’ complex. It is possible that the close arrangement of the sulphur atoms contributes to the global repulsion of the two proteins, leading to their dissociation.

## 3. Discussion

Given the important role of the endoplasmic reticulum-resident transmembrane protein hVKORC1 in several vital physiological and homeostasis processes, detailed knowledge of each step of its life cycle is necessary to improve our understanding of hVKORC1 functions. The use of hVKORC1 as a therapeutic target in the prevention and treatment of various blood, heart, and cancer diseases [23] has stimulated numerous empirical and theoretical studies of its catalytic activity, which culminated in the development of competitive vitamin K inhibitors, the vitamin K antagonists (VKAs). However, numerous mutations of hVKORC1 provoke modifications of its activity [24] and induce hypersensitivity [25] or resistance phenomena to VKAs [26,27]. Therefore, the search for alternative ways to regulate or modulate the functioning of this enzyme is required. Thus, researchers focusing on VKORC1 problems have addressed both fundamental and applied questions.

By focusing on the activation process of hVKORC1 by its redox protein, a less studied step in the hVKORC1 life cycle, we pursued this research to enrich the knowledge (or rather close the gap) associated with 3D models of different states, which occur during thiol–sulphide exchange reactions between the reduced PDI and oxidised hVKORC1. In particular, a more extensive MD simulation of the ‘precursor’ complex formed prior to the initiation of TDSE reactions confirmed our preliminary conclusion for the preferential spatial orientation of PDI relative to hVKORC1 and, accordingly, the correctness of the model of this state [16,18]. By exploring possible models that could describe the ‘intermediate’ covalently bound intermolecular complex, we postulated that the best solution was obtained for a model with the same PDI orientation as the non-covalently bound ‘precursor’ complex.

This model describes an ‘intermediate’ state of the PDI-hVKORC1 complex, in which the PDI cysteine residue C37 binds to C51 of hVKORC1, forming a ‘mixed’ covalently bound macromolecular complex. TDSE reactions culminate in the transfer of both protons from PDI to hVKORC1, thereby stabilising the new state—the ‘successor’ complex. This final step of the intermolecular oxidation–reduction process is represented by a non-covalently bound complex, followed by the complete dissociation of two proteins—oxidised PDI and reduced hVKORC1. Thus, the three reversible states of the PDI-VKORC1 complex are represented by three 3D models, which plausibly mimic the functional states of this pair of proteins.

For proteins or their molecular complexes, the stable conformers captured experimentally represent a functional cycle’s initial or final states. In contrast, short-lived or highly flexible transient intermediates, which are often key to understanding molecular mechanisms, are challenging to capture. We believe, and even argue, that our proposed models that mimic functional states of the oxidation–reduction process between PDI and hVKORC1 will be useful for basic or applied future research.

What do these models provide and how can they be used? First, experimental confirmation is highly desirable for their use in other cutting-edge studies. However, even without such approval, these models, supported by statistical approaches, can be used to study spontaneous intermolecular reactions of thiol–disulphide exchange by the hybrid quantum-mechanics/molecular mechanics (QM/MM) simulation, which will be very useful for investigating de novo disulphide bond formation and thiol–disulphide exchange. Such a coupled simulation could potentially account for both the kinetics and thermodynamics of the exchange process, but also have the highest computational costs.

In parallel, this process can be studied using force-clamp MD simulations coupled with an energy-based exchange (swapping) criterion to simulate the dynamics of force-induced unfolding while allowing disulphide shuffling [28]. This purely classical approach has unique advantages over quantum-mechanics (QM) methods [29,30]: it does not require an a priori determination of the reacting residues. However, it can instead predict them and take into account large-scale protein movements and their interactions with the reaction at the atomistic level. Moreover, the results of this method will either validate or refute our proposed models.

These convoluted studies, in turn, could encourage the classification of disulphides into a catalytic or allosteric type [31]. An allosteric disulphide bond constitutes an ‘allosteric site’ if the conformational changes caused by breaking the chemical bond alter the proteins’ function. Such research may be useful for understanding how enzymes tune redox cofactors and recruit oxidants to improve the specificity and efficiency of disulphide formation.

In PDI-hVKORC1 complexes, we observed conformational changes and alternating folding in both interacting partners, PDI and hVKORC1. The hVKORC1 L-loop showed an open-to-closed shape transition and an increase in the helical fold in all PDI-hVKORC1 complexes compared to unbound hVKORC1. In turn, in PDIs that interact with hVKORC1, the αH2 helix containing the CGHC motif decreases its fold relative to the unbound protein. Such observations illustrate the adaptability of these two intrinsically disordered fragments, the L-loop of hVKORC1 and the N-terminus of the αH2-helix of PDI, resulting in an increased specificity and efficiency of molecular recognition between these proteins required for the thiol–disulphide exchange reactions, and likely indicating an allosteric regulation of this process.

Interestingly, considerable folding modifications of these two fragments in PDI-hVKORC1 complexes do not significantly affect the non-covalent interactions of R61 (hVKORC1) with A42 and E43, which are the main factors maintaining non-covalently bound complexes.

Because the intermolecular interface formed by non-covalent bonds is globally conserved during thiol–disulphide exchange reactions, it can be used as a target for the development of inhibitors that act as modulators of PDI-hVKORC1 interactions. Together with the targeting of allosteric pockets found in the L-loop of hVKORC1 [32], this interface can be identified as a new target that can be used for developing ‘allo-network’ modulators [33]. This innovative concept is based on two types of inhibition: intra-protein competitive or allosteric inhibitors and inter-protein modulators interacting at the protein–protein interaction interface. Such an approach is a way to improve treatment by increasing the drugs’ specificity, avoiding or significantly reducing the side-effects caused by non-specific molecules, and possibly limiting the rapid evolution of new protein strains.

The results presented here are novel and original, and despite the lack of experimental confirmation, they are undoubtedly of interest to both fundamental and applied research.

## 4. Materials and Methods

### 4.1. 3D Models

***PDI-hVKORC1***. The coordinates of two structural models of the full-length PDI-hVKORC1 ‘precursor’ complex, Model I-1 and Model I-2, generated by using two crystallographic structures of human PDI (PDB ID: 4EKZ) and bacterial VKOR (PDB ID: 4NV5) [20], were taken from [16] and used for a more extended (0.52 μs) MD simulation. Taking the MD conformations of the PDI-hVKORC1 ‘precursor’ complex models, at 441 and 486 ns for MI-1 and MI-2, respectively, as templates, other models of PDI-hVKORC1 were constructed by converting thiol groups (–SH) to a disulphide bond (–S-S–) and vice versa. Because the thiol group of PDI can interact with either C41 or C53 of hVKORC1 to form a ‘mixed’ –S-S– bond, the modelling of the ‘intermediate’ complex (state II) is represented by a pair of models derived from each model of the ‘precursor ‘complex. These models are labelled as Model II-11, Model II-12, Model II-21, and Model II-22. The ‘successor’ complex describing the final state stabilised after TDSE reactions is referred to as Model III-1. Each model of the ‘intermediate’ and ‘successor’ complex of PDI-hVKORC1 was built and optimised using Modeller [34]. The optimisation of each model was repeated twice with an objective function cut-off of 10^6^.

The stereochemical quality of all 3D models was assessed by Procheck [35], which revealed that more than 96–98% of the non-glycine/non-proline residues have dihedral angles in the most favoured and permitted regions of the Ramachandran plot, as is expected for good models.

### 4.2. Molecular Dynamics Simulation

#### 4.2.1. Preparation of the Systems 

For MD simulations, all models of the PDI-VKORC1 complexes were prepared with the LEAP module of AMBER 16 [36] using the *ff14SB* all-atom force field parameter set [37]: (i) hydrogen atoms were added; (ii) covalent bond orders were assigned; (iii) protonation states of amino acids were assigned based on their solution for pK values at neutral pH, and histidine residues were considered neutral and were protonated for ε-nitrogen atoms; and (vi) a Na^+^ counter-ion was added to neutralise the protein charge.

Each model was embedded in the equilibrated and hydrated membrane composed of 200 1,2-dilauroyl-sn-glycero-3-phosphocho-line (DLPC) lipids using the replacement method available in the CHARMM-GUI Membrane Builder [38]. This lipid bilayer had been completed with water molecules (TIP3P) [39] and pre-equilibrated during 1.5 ns of MD using the *Lipid14* tool [40] from the AMBER package.

Each protein complex inserted into a membrane was solvated with explicit TIP3P water molecules in a periodic rectangular box with a distance of at least 12 Å between the proteins and the boundary of the water box. Cl^−^ ions were randomly placed to neutralise the system.

#### 4.2.2. Set-Up of the Systems

The set-up of the systems was performed with the SANDER module [41] of AMBER18. First, each system was minimised successively using the steepest descent and conjugate gradient algorithms as follows: (i) 10,000 minimisation steps where the water molecules have fixed protein atoms, (ii) 10,000 minimisation steps where the protein backbone is fixed to allow protein side-chains to relax, and (iii) 10,000 minimisation steps without any constraint on the system. The equilibration was performed on the solvent, keeping the solute atoms (except H-atoms) restrained for 100 ps at 310 K and a constant volume (NVT). The protein, membrane, and solvent (water and ions) temperatures were separately coupled to the velocity rescale thermostat, which was a modified Berendsen thermostat [42] with a relaxation time of 0.1 ps. Each system was equilibrated during 1 ns (NPT) with all non-hydrogen atoms of the protein and the DLPC membrane harmonically restrained. A semi-isotropic coordinate scaling and Parrinello–Rahman pressure coupling were used to maintain the pressure at 1 bar, with a relaxation time of 5 ps. The Nose–Hoover thermostat [43] was applied to the protein, lipids, and solvent (water and ions) separately, with a relaxation time of 0.5 ps to keep the temperature constant at 310 K. Water and ions were allowed to move freely during the equilibration.

#### 4.2.3. Production of the MD Trajectories 

All trajectories were performed using the AMBER ff14SB force field with the PMEMD module of AMBER 16 and AMBER 18 [36] (GPU-accelerated versions) running on a local hybrid server (Ubuntu, LTS 14.04, 252 GB RAM, 2× CPU Intel Xeon E5-2680, and Nvidia GTX 780ti) and on the supercomputer JEAN ZAY at IDRIS.

Each fully relaxed PDI-hVKORC1 complex inserted into the solvated bilayer lipid membrane was simulated during the 0.52 µs MD trajectory. A time step of 2 fs was used to integrate the equations of motion based on the Leap-Frog algorithm [44]. Coordinate files were recorded every 1 ps. Neighbour searching was performed by the Verlet algorithm [45]. The Particle Mesh Ewald (PME) method [46] with a cut-off of 9.0 Å was used to treat long-range electrostatic interactions at every time step. The van der Waals interactions were modelled using a 6–12 Lennard–Jones potential. The initial velocities were reassigned according to the Maxwell–Boltzmann distribution.

### 4.3. Data Analysis

Unless otherwise stated, all recorded MD trajectories were analysed with the standard routines CPPTRAJ 4.15.0 program [47] of AMBER 18 Suite. The RSMD and RMSF values were calculated for the Cα-atoms using the initial model (at t = 0 ns) as a reference. All analysis was performed on the MD conformations considering either all simulations or the production part of the simulation, which was generated after the removal of non-well-equilibrated conformations as was shown by the RMSDs, or on residues with a fluctuation of less than 4 Å as shown by the RMSF. In particular, for the characterisation of PDI-hVKORC1 complexes, the residues from 6 to 156 of hVKORC1 and from 1 to 111 of PDI were used.

**Secondary structure.** The secondary structural propensities for all residues were calculated using the Define Secondary Structure of Proteins (DSSP) method [48]. The secondary structure types were assigned for residues based on backbone -NH and -CO atom positions. Secondary structures were assigned every 20 ps for each trajectory.

**Non-covalent distance monitoring.** The H-bonds between heavy atoms (N, O, and S) as potential donors/acceptors were calculated with the geometric criteria: donor/acceptor distance cut-off was set to 3.6 Å, and the bond angle cut-off was set to 120°. Hydrophobic contacts were considered for all hydrophobic residues with side chains within a distance of 4 Å of each other.

**Free energy calculation.** The molecular mechanics (MM) energies were calculated from the last stable 300 ns trajectories with the generalised Born and surface area continuum solvation (MM/GBSA) method [49,50] implemented in AmberTools18. The tool ‘*g_mmpbsa*’, which calculates the binding energy of biomolecular associations like protein–protein, was utilised for potential energy and solvation free energy calculations (with default parameters) [22].

**Graphics**. Visual inspection of the conformations and figure preparation was made with PyMOL [51]. The VMD 1.9.3 program [52] was used to prepare the protein MD animations. To visualise the motions along the principal components, the Normal Mode Wizard (NMWiz) plugin [53], which is distributed with VMD, was utilised.

## Figures and Tables

**Figure 1 ijms-25-04135-f001:**
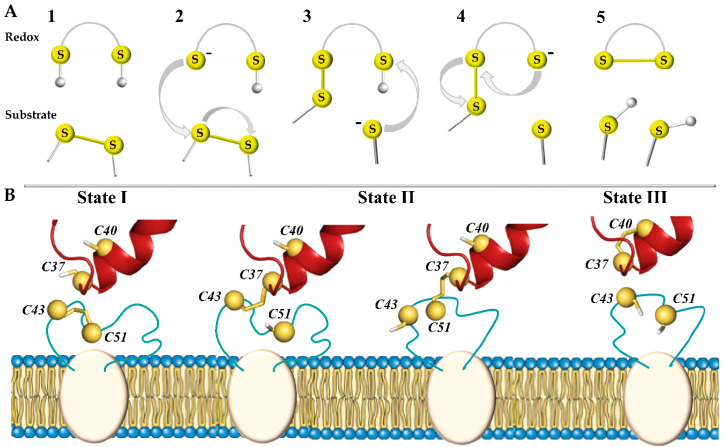
Thiol–disulphide exchange (TDSE) reactions triggered by the redox protein (PDI) regulate disulphide bond reduction in proteins (hVKORC1). (**A**) In the initial step (**1**), the CXXC motif of the redox protein is protonated and the substrate contains a disulphide bond. If N-terminal cysteine in the redox motif is deprotonated and takes the form of a thiolate anion (–S–) (**2**), this potent nucleophile interacts with the substrate and allows the formation of a mixed disulphide bond between the two proteins (**3**). During this reaction, conformational changes allow deprotonation of the second cysteine from the CXXC motif, releasing the trapped mixed disulphide (**4**) and leading to the formation of a disulphide bond in the substrate protein (**5**). (**B**) Three states of the PDI-hVKORC1 pair—‘precursor, ‘intermediate’, and ‘successor’ complexes describing steps **1**, **3**, and **5** of the thiol redox reactions, respectively. TDSE reactions are considered only for the intermolecular interactions of the PDI-hVKORC1 protein pair, but not for the intramolecular process of hVKOCR1.

**Figure 2 ijms-25-04135-f002:**
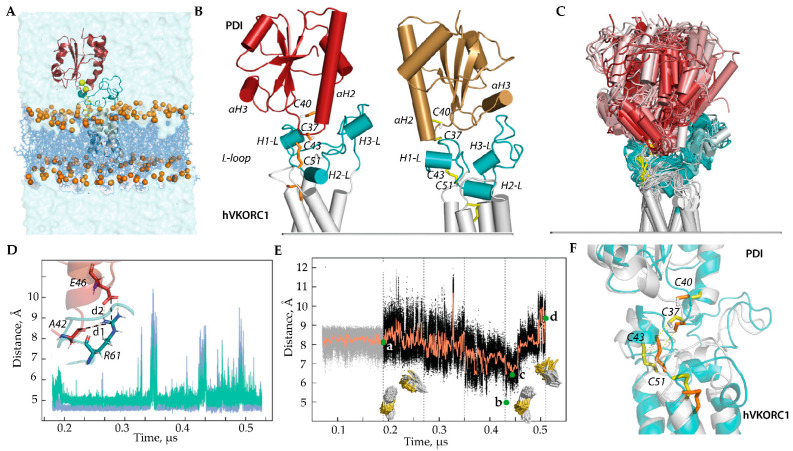
Modelling of the ‘precursor’ complex PDI-hVKORC1 in state I. (**A**) PDI-hVKORC1 complex embedded in a membrane and surrounded by water. (**B**) MD conformations of Model I-1 (**left**) and Model I-2 (**right**) taken at t = 0.18 µs. (**C**) Superimposition of MD conformations of Model I-1 taken every 20 ns. The colour gradient shows time-dependent conformations from light (t = 0) to dark (t = 5.2 μs). Proteins are shown as a cartoon: PDI, L-loop, and TMD of hVKORC1 are in red, cyan, and grey, respectively. The thiol group of cysteine residues and the S-S bond are shown as yellow sticks. (**D**) Distances d1 (blue) and d2 (green) between R61 (hVKORC1) and E46, and between R61 and A42 (PDI) (shown in the insert) observed during MD simulation of Model I-1. (**E**) Variations in distance (black) and their mean values (orange) between sulphur atoms from cysteine residues C37 (PDI) and C43 (hVKORC1) observed in the MD simulation of Model I-1. Green dots correspond to the distance S⋯S in the first conformation after the long-time equilibrium (**a**), with a minimal value (low population) (**b**), with an ‘optimal’ value (average population) (**d**), and in the last frame (**c**), where d = 8.12, 4.97, 6.40, and 9.36 Å, respectively. The insets show the orientation of the C37 thiol group (PDI) relative to the C43–C51 disulphide bond of hVKORC1 at time intervals of 180–350 ns (**left**) and 350–520 ns (**right**). (**F**) Two MD conformations, taken at t = 320 ns (grey, yellow) and t = 441 ns (teal, orange), represent different orientations of the thiol group from C37 of PDI.

**Figure 3 ijms-25-04135-f003:**
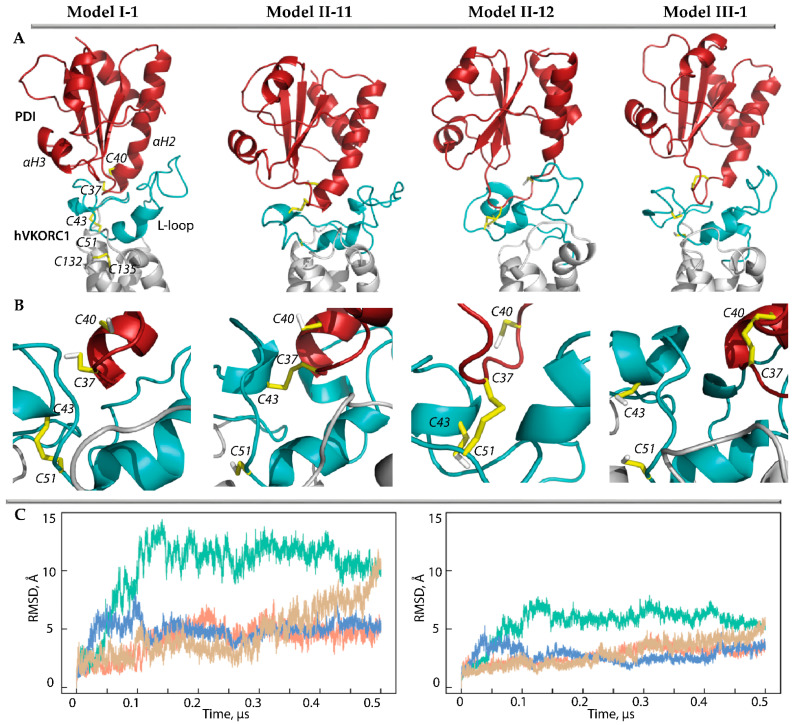
Modelling of PDI-hVKORC1 complexes assembled during PCET reactions. (**A**,**B**) Three-dimensional models of the ‘precursor’ complex (Model I-1) taken as the starting structure, two possible ‘intermediate’ models with covalently linked cysteine residues C37 and C43 (Model II-11), C37 and C51 (Model II-12), and ‘successor’ complex (Model III-1). Each model is represented by MD conformations taken at 50 ns of MD trajectories. The proteins are shown as a cartoon: PDI, L-loop, and TMD of hVKORC1 are in red, cyan, and grey, respectively. The thiol group of cysteine residues and the –S-S– bond are shown as yellow sticks. (**C**) RMSDs from the initial coordinates calculated for the Cα-atoms in the PDI-hVKORC1 complexes (**left**) and L-loop (**right**) after fitting to initial conformation (t = 0 µs). The RMSD curves characterising models MI-1, MII-11, M II-12, and M III-1 are coloured in orange, green, blue, and sand, respectively.

**Figure 4 ijms-25-04135-f004:**
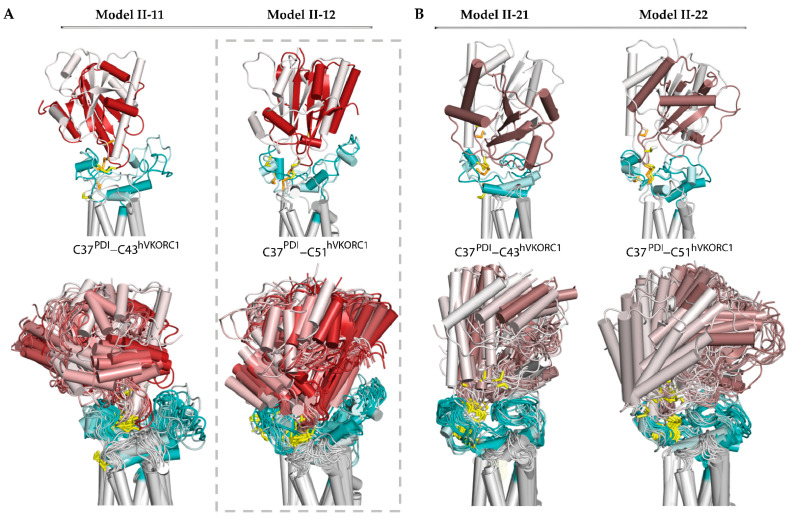
Modelling of the ‘intermediate’ PDI-hVKORC1 complex (state II) formed by the PDI cysteine residue C37 covalently bonded with C43 or C51 of hVKORC1. (**A**) Two possible models, Model II-11 and Model II-22, generated from the ‘precursor’ complex (Model I-1). (**B**) Two possible models, Model II-21 and Model II-12, generated from the ‘precursor’ complex (Model I-2). (**A**,**B**) Superimposition of MD conformations of each model taken at t = 0 and t= 0.52 µs (top) and every 20 ns (bottom). The colour gradient shows time-dependent conformations from light (t = 0) to dark (t = 0.52 μs). Proteins are shown as coloured cartoons: the PDI, L-loop, and TMD of hVKORC1 are displayed in red/brown, cyan, and grey, respectively. The thiol group of cysteine residues and the S-S bond are shown as yellow sticks.

## Data Availability

The data presented in this study are available on request from the corresponding author.

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
