# Peer review of "Vitamin K Epoxide Reductase Complex–Protein Disulphide Isomerase Assemblies in the Thiol–Disulphide Exchange Reactions: Portrayal of Precursor-to-Successor Complexes"

_ijms, 2024, doi:10.3390/ijms25084135_

Round 1
Reviewer 1 Report
Comments and Suggestions for Authors
Please see the attachment.

Author Response
Reviewer 1
Comments and Suggestions for Authors
Stolyarchuk et al. presented a computational work providing the modeling of the disulphide
isomerase (PDI) in complex with the human vitamin K epoxide reductase complex 1 (hVKORC1), because of PDI may be a physiological redox partner of hVKORC1. Thus, the Authors provided the models for the precursor, the intermediate and the ‘successor’ states of PDI-hVKORC1 complex.
MD simulations of about 500ns were performed for each complex and standard analysis
were performed to characterize a.a. residue orientation and energy profiles.
Response: The Authors thank Reviewer 1 for the comments on the manuscript and critical remarks that were considered in the revised version or explained below.
I suggest the acceptance of the paper after minor revision:
- Fig B Model I-1 and I-2 should be superimposed and then represented separately in panel to give the same reference point to the observer:
Response: Since two models of the hVKORC1-PDI precursor complex were the focus of two previous works (Stolyarchuk et al., 2021; Ledoux et al., 2022), a figure showing their superimposition was published in Suppl. Material. We are unable to re-publish this material. To better explain the differences between the models, we added the labels for the reference helices (Fig. 2). The fundamental difference between the models is the position of the PDI relative to hVKORC1 (rotation by 180°).
- Fig 3A the Authors should label all the figures in the panel (e.g. Model II-11, Model
II-12 etc..)
Response: The Authors thank Reviewer 1 for this comment. The lines in Figure 3 separate the panels to avoid labelling repeats.
- Apart from the figures, I am curious about the modeling of the intermediate state that should have the thiolate anion. How did the Authors treat the presence of the S- ? Maybe the Authors could add just a sentence to show that they have thought about it. To be noted, I have found this paper online that may be of interest for the Authors: Izmailov, S.A., Podkorytov, I.S. & Skrynnikov, N.R. Simple MD-based model for oxidative folding of peptides and proteins. Sci Rep 7, 9293 (2017). https://doi.org/10.1038/s41598-017-09229-7
Response: The Authors thank Reviewer 1 for this comment. Our manuscript did not consider states containing the thiolate (–S-) in PDI or hVKORC1. Only the state representing the intermediate covalent complex in which C37 of PDI forms the disulphide bond with C43 or C51 of hVKORC1 was modelled. In these two alternative models, the second cysteine of each protein is in the protonated state (see Figure 1). In this regard, the authors of the recommended article introduced “cysteine residues which contain deprotonated thiols, but remain overall neutral”. I see a problem here. It is well known that parameters for modelling thiolates using molecular mechanical force fields have not yet been validated, partly due to the lack of structural data on thiolate solvation. However, it has been shown by physicists (E. Awoonor-Williams & Ch. Rowley, 2018) the importance of the definition of distinct unbound parameters for the protonated/deprotonated states of amino acid side chains in MM force fields.

Reviewer 2 Report
Comments and Suggestions for Authors
ijms-2928208
The human vitamin K epoxide reductase complex (hVKORC1) plays an important role in blood clotting. hVKORC1 in the reduced form catalyzed the reduction of vitamin K epoxide, and it changes to oxidized form. Intensive studies have been done to reveal the reactivation mechanism of hVKORC1. The authors recently proposed, using in silico methods, that PDI is the protein most compatible with hVKORC1 compared to the thioredoxin-like proteins and is most likely its redox protein. The idea is supported by the recent study that verexpression of PDI enhances vitamin K epoxide reductase activity (doi: 10.1139/bcb-2021-0441). The authors have conducted in silico studies to reveal the details mechanism for thiol-disulphide exchange reactions of hVKORC1-PDI. The MD simulations are well performed to show the ‘precursor’, ‘intermediate’, and ‘successor’ states of the thiol-disulphide exchange reactions. Although, it is difficult to obtain experimental data to support the model, the results are informative for the further study. The authors also propose using these complexes to develop of ‘allo-network drugs’ necessary for treating blood diseases. However, the discussion on it is too limited in the manuscript. I would like to request the authors to give some more comments on the development of ‘allo-network drugs’ before accepting this manuscript.
Author Response
Reviewer 2
Comments and Suggestions for Authors
The human vitamin K epoxide reductase complex (hVKORC1) plays an important role in blood clotting. hVKORC1 in the reduced form catalyzed the reduction of vitamin K epoxide, and it changes to oxidized form. Intensive studies have been done to reveal the reactivation mechanism of hVKORC1. The authors recently proposed, using in silico methods, that PDI is the protein most compatible with hVKORC1 compared to the thioredoxin-like proteins and is most likely its redox protein. The idea is supported by the recent study that verexpression of PDI enhances vitamin K epoxide reductase activity (doi: 10.1139/bcb-2021-0441). The authors have conducted in silico studies to reveal the details mechanism for thiol-disulphide exchange reactions of hVKORC1-PDI. The MD simulations are well performed to show the ‘precursor’, ‘intermediate’, and ‘successor’ states of the thiol-disulphide exchange reactions. Although, it is difficult to obtain experimental data to support the model, the results are informative for the further study. The authors also propose using these complexes to develop of ‘allo-network drugs’ necessary for treating blood diseases.
Response: The Authors thank Reviewer 2 for the comments on the manuscript and critical remarks that were considered in the revised version of the manuscript or explained below.
However, the discussion on it is too limited in the manuscript. I would like to request the authors to give some more comments on the development of ‘allo-network drugs’ before accepting this manuscript.
Response: This novel strategy was discussed in our previous paper (M. Botnari & L. Tchertanov ). “This innovative concept of “allo-network drugs” is based on two types of inhibition: intra-protein competitive or allosteric inhibitors and inter-protein modulators interacting at the protein-protein interaction interface [26]. Such an approach is a way to improve treatment by increasing the drugs’ specificity, avoiding or significantly reducing the side effects caused by non-specific molecules, and possibly limiting the rapid evolution of new protein strains”. I added these two sentences to the present manuscript to avoid incomprehension.

Reviewer 3 Report
Comments and Suggestions for Authors
This is a good paper that reports and MD study thiol-disulfide exchange between Vitamin K Epoxide Reductase Complex and protein disulfide isomerase. The models obtained can be a basis for QM or QM/MM studies of the detailed mechanism. The calculations and analysis have been performed correctly and the conclusions are sound. I have minor suggestions:
1. Although the authors refer to their former paper (ref 16) as a source of the structures which served to construct the initial models, at least the PDB codes of the two proteins should be quoted.
2. The beginning of te legend of Figure 1 is unclear and should be revised:
"Thiol-disulphide exchange (TDSE) reactions triggered by protein redox protein (PDI) reg-ulate disulphide bond reduction in proteins (hVKORC1)."
Probably "protein redex protein" should be "the redox protein".
3. page 3, lines 11-13:
"Identification of the physiological redox partner of hVKORC1 has been the subject of numerous in vitro and in vivo studies for an extended period and remains valid today [14, 15]."
Probably not "remains valid today" but "is still being investigated" or so.
4. page 4, lines 16-17: "for inter-protein distance S•••S"
Probably "for the S•••S inter-protein distance".
5. page 6, lines 3-4 from the bottom:
"Overall, the L-loop helical fold is slightly increased (by 10%) in Model II-12 compared to Model II-11."
I guess that the authors mean the increase of rmsd of this fragment from that of the reference structure. If they mean loop dimension (diameter, radius of gyration, etc.), they should specify this measure.
Author Response
Reviewer 3
Comments and Suggestions for Authors
This is a good paper that reports and MD study thiol-disulfide exchange between Vitamin K Epoxide Reductase Complex and protein disulfide isomerase. The models obtained can be a basis for QM or QM/MM studies of the detailed mechanism. The calculations and analysis have been performed correctly and the conclusions are sound.
Response: The Authors thank Reviewer 3 for the positive comments on the manuscript and critical remarks that were considered in the revised version of the manuscript or explained below.
I have minor suggestions:
- Although the authors refer to their former paper (ref 16) as a source of the structures which served to construct the initial models, at least the PDB codes of the two proteins should be quoted.
Response: The reference codes (PDB ID) were added in the Methods.
- The beginning of the legend of Figure 1 is unclear and should be revised:
"Thiol-disulphide exchange (TDSE) reactions triggered by protein redox protein (PDI) reg-ulate disulphide bond reduction in proteins (hVKORC1)."
Probably "protein redex protein" should be "the redox protein".
Response: The legend was corrected.
- page 3, lines 11-13:
"Identification of the physiological redox partner of hVKORC1 has been the subject of numerous in vitro and in vivo studies for an extended period and remains valid today [14, 15]."
Probably not "remains valid today" but "is still being investigated" or so.
Response: Thank you for the better version. It was modified.
- page 4, lines 16-17: "for inter-protein distance S•••S"
Probably "for the S•••S inter-protein distance".
Response: Thank you for the better version. It was modified.
- page 6, lines 3-4 from the bottom:
"Overall, the L-loop helical fold is slightly increased (by 10%) in Model II-12 compared to Model II-11."
I guess that the authors mean the increase of rmsd of this fragment from that of the reference structure. If they mean loop dimension (diameter, radius of gyration, etc.), they should specify this measure.
Response: Thank you for the remark. The helical fold estimation was performed using the secondary structure interpretation (DSSP, see Methods and Suppl. Mat).
